# Effects of Group-I Elements on Output Voltage Generation of ZnO Nanowires Based Nanogenerator; Degradation of Screening Effects by Oxidation of Nanowires

**DOI:** 10.3390/mi13091450

**Published:** 2022-09-01

**Authors:** Mansoor Ahmad, M. K. Ahmad, M. H. Mamat, A. Mohamed, A. B. Suriani, N. M. A. N. Ismail, C. F. Soon, N. Nafarizal

**Affiliations:** 1Microelectronic and Nanotechnology—Shamsuddin Research Centre (MiNT-SRC), Faculty of Electrical and Electronic Engineering, Universiti Tun Hussein Onn Malaysia (UTHM), Parit Raja 86400, Malaysia; 2NANO-ElecTronic Centre (NET), School of Electrical Engineering, College of Engineering, Universiti Teknologi MARA, Shah Alam 40450, Malaysia; 3Nanotechnology Research Centre, Department of Physic, Faculty of Science and Mathematics, Universiti Pendidikan Sultan Idris, Tanjung Malim 35900, Malaysia

**Keywords:** ZnO nanowires, piezoelectric potential, VING, Schottky contact, nanogenerator

## Abstract

Here, we report the successful incorporation of group I elements (K, Na, Li) to ZnO nanowires. Three distinct (2, 4, and 6 wt.%) doping concentrations of group I elements have been used to generate high piezoelectric voltage by employing a vertically integrated nanowire generator (VING) structure. X-ray photoelectron spectra (XPS) indicated the seepage of dopants in ZnO nanowires by substitution of Zn. Shallow acceptor levels (Li_Zn_, Na_Zn_, K_Zn_) worked as electron trapping centers for intrinsically *n-type* ZnO nanowires. Free moving electrons caused a leakage current through the nanowires and depleted their piezoelectric potential. Reverse leakage current is a negative factor for piezoelectric nanogenerators. A reduction in reverse leakage current signifies the rise in output voltage. A gradual rise in output voltage has been witnessed which was in accordance with various doping concentrations. K-doped ZnO nanowires have generated voltages of 0.85 V, 1.48 V, and 1.95 V. For Na-doped ZnO nanowires, the voltages were 1.23 V, 1.73 V, and 2.34 V and the voltages yeilded for Li-doped ZnO nanowires were 1.87 V, 2.63 V, and 3.54 V, respectively. Maximum voltage range has been further enhanced by the surface enrichment (oxidized with O_2_ molecules) of ZnO nanowires. Technique has been opted to mitigate the screening effect during an external stress. After 5 h of oxidation in a sealed chamber at 100 ppm, maximum voltage peaks were pronounced to 2.48 V, 3.19 V, and 4.57 V for K, Na, and Li, respectively. A low-cost, high performance mechanical transducer is proposed for self-powered devices.

## 1. Introduction

ZnO is one of the most promising group II–VI semiconductor materials, having a wide band gap of 3.36 eV and large exciton binding energy of 60 meV [1] due to which it is considered as an ideal choice to be used in various applications such as light emitting diodes [2], gas sensors [3,4,5], ultraviolet lasers [6], solar cells [7], photodetector [8], and nanogenerators [9,10]. The pertinent reason for researchers to choose ZnO as a base material to fabricate Nano-scale devices is due to its easy and low cost growth techniques [11,12]. The reproducibility of single crystal structure at Nano-scale level is quite challenging sometimes and the major reason why ZnO has gained the attention of researchers is that various morphologies such as nanobelts [13], nanosprings [14], nanocombs [15], nanowires [16], etc. could be grown efficiently under ambient conditions. ZnO nanowire-based piezoelectric nanogenerators [17,18,19] have been the core area of our research and this recent attempt is a continuation of our endeavors in this area. ZnO in its wurtzite structure exhibits a lack of any central symmetry. Under normal (unstrained) conditions, Zn^+2^ (cations) and O^−2^ (anions) are stacked one over the other in a layer-by-layer fashion, and the charge center of the cations and anions coincide with each other. As the crystal is slightly perturbed by an external force, the charge symmetry disrupts and cations and anions move apart in the crystal. The separation of anions and cations generates an electric-dipole in the crystal and the accumulation of charges on either side of the crystal develops a potential inside the structure which is termed as piezoelectric potential [7,8,11]. If the crystal is connected to an external load, developed piezoelectric potential could be delivered at the output stages and there would a transient flow of electrons through the external circuit. So, the working principle of the piezoelectric nanogenerators is based on piezopotential developed within the ZnO crystal by minute external force. For the generation of high output voltage, Schottky contact is the other key factor; metal semiconductor contact is quite vital in this case since it ensures the flow of the electrons in one direction only (i.e., through an external circuit). The work function for the metal must be less than electron affinity of the semiconductor. Au and Pt work have higher work functions than the electron affinity of ZnO, so both metals are ideal choices for a Schottky contacts [20,21].

In this study, we have used a two step approach to enhance the piezoelectric output voltage, first by *p-type* doping from group I elements (Na, K, Li) to pristine ZnO nanowires and secondly by oxidizing doped ZnO nanowires with O_2_ molecules to reduce the screening effects. ZnO nanowires doped with *p-type* impurities have not been a favorite choice of the researchers due to their poor stability at room temperature and converting back to *n-type* conductivity [22,23]. Primarily, this may be due to their intrinsic donor defects, but there are few reports which indicated the formation of shallow acceptor levels that contributed towards the *p-type* conductivity in ZnO nanowires [24]. The ionic radii of Li (0.76 Å) is smallest in group I elements and comparable to Zn (0.74 Å) so Li ions can easily move inside ZnO crystal and occupy interstitials, Na and K having an ionic radii 1.02 Å and 1.38 Å, respectively. However, both are cations and, in general, cations have smaller ionic radii than parent element. As Na has 11 electrons and three energy levels, effective nuclear charge on the valance electron is +1 since the outer valance electron is being protected by 10 core electrons and 11 protons in the nucleus (so (11 − 10 = +1)), but Na^+1^ ion has ten electrons since one valance electron has been removed; thus, it has two energy levels (2 electrons, 8 electrons in 1st and 2nd energy levels, respectively) while the Na element has three energy levels (2, 8, 1 electrons in three energy levels, respectively), so one energy level is reduced in addition to the ionic radii [25,26]. Both Na and Li are supposed to enter ZnO crystal interstitially to form shallow acceptor levels. However, theoretical studies suggest that Li can inter ZnO nanowire either by substituting Zn ion Li_Zn_ or occupying interstitial positions (Li_i_). Similarly, K has 19 electrons while K^+1^ ion has 18 electrons. The ionic radius of K^+^ is shrunk as the number of electrons has reduced, and this means repulsive force between electrons is reduced (and so is their ionic radius) [27]. A recent study has been envisioned to generate high output voltage by adding minute doping concentrations of Na, K, and Li to ZnO nanowires. These acceptor impurities are supposed to capture free electrons and can reduce leakage current through nanowires. A low cost and easy aqueous route has been chosen to incorporate the dopant in ZnO nanowires since Ex-situ doping techniques such as ion implantation [28], pulsed lase deposition [29], and thermal diffusion [30] are considered more damaging to the crystal structure.

## 2. Material and Methods

All reagents used were of analytical grade with 98% purity. ITO (indium tin oxide) coated PET substrate (poly ethylene terephthalate) worked as bottom electrode, Au sputtered gold electrode as top electrode and *p-type* doped ZnO nanowires are the intermediate part of VING. Substrates were cleaned by deionized water and acetone, respectively, in an ultrasonic bath for 15 min. each. 10 mM solution of Zinc acetate dihydrate [Zn(CH_3_COO)_2_⋅2H_2_O] has been used to grow a seed layer on ITO coated PET substrates. In second step, nutrient solution comprised of hexamethylenetetramine [C_6_H_12_N_4_] and Zinc nitrate hexahydrate [Zn(NO_3_)_2_.6H_2_O] has been prepared for the growth of ZnO nanowires. ITO coated PET substrates has been placed upside down in nutrient solution for 3 h at 90 °C. The concentration of nutrient solution has been carefully adjusted by using the following relation
C = [m/V] × [1/MV] (1)
where C is the molar concentration or molarity of the solution, m is the mass of solute in grams V is the volume of the solution in which solute needs to dissolved and MV is the molecular weight in g/mol. It has been explained earlier [4,5] that the density of the ZnO nanowires were closely related with the concentration of nutrient solution and morphology of nanowires was dependent on growth time and temperature. Emscope SC500 is used to sputter Au electrode on top of ZnO nanowires, Ar gas with a pressure 0.1 Torr is being filled in the chamber, 2 kv operating voltage is being used to initiate charge irradiation inside the chamber. For K-doped ZnO nanowires K_x_Zn_1−x_O (x = 0%, 2%, 4%, 6% and 10%) have been prepared by using K (powder form) has been directly mixed in the nutrient solution. nutrient solution was kept at 90 °C for 4h. For Li- doped ZnO nanowire lithium acetate dihydrate (LiCH_3_COO·2H_2_O) of analytical grade (98% purity) has been used. Minute Li concentrations of 2, 4, and 6 wt.%. have been obtained by adjusting the values mentioned in relation (Equation (1)). The nutrient solution is stirred for 30 min. at 60 °C. For Na-doped ZnO NaF was dissolved in de-ionized water and solution was stirred for 30 min at 50 °C for. Zn(OH)_2_ being the byproduct was filtered many times during the procedure. Na-doped concentration was varied from 2 to 10 wt.% but for uniformity only 2, 4 and 6 wt.% has been further analyzed. XPS studies have been performed by using PHI 5000 scanning ESCA microprobe using (hv = 1486 eV) and X-ray source was operated at 15 KV, power 25 W, 100 µm spot size. XRD pattern have been obtained by Shimadzu—6000 X-ray diffractometer having source Cu Kα (λ = 1.54 Å). The schematic diagram of VING has been shown in Figure 1. Picoscope 5204 has been used to measure output voltages generated by the structure. VING structure have been further oxidized with O_2_ to mitigate the screening effects during the external stress. VING has been oxidized for 5 h in a sealed chamber at 100 ppm to obtain maximum output voltage. The oxidation phenomenon has been very well explained in our earlier reports [15,16,17,18], which has significantly reduced the screening effect during external compression.

## 3. Results and Discussions

Surface morphologies of doped and un-doped ZnO nanowires have been examined by ESEM images by using Philips XL30. Figure 2a shows the topographical image of as-grown ZnO nanowires which ensures the vertical growth orientation of nanowires. Figure 2b–d also shown the same trend. Minute dopants have not altered the perpendicular growth orientation but a slight variation in diameter range has been observed. SEM images clearly indicated the uniform growth of ZnO nanowires on PET substrate. To examine the diameter distribution of doped and un-doped ZnO nanowires Image J software has been used [31]. Initially, particular SEM image has been selected to measure top surface area of ZnO nanowires. Selected image has been replicated in image J software then converted it in 8-bit image. In the second step, calibration of scale has been carried out, as image J doesn’t understand the SEM image scale. In the third step, a grey image has been obtained by using binarization technique and in the final step, by adjusting the contrast of the modified image, only the top surface area of nanowires has been selected (the rest a discarded). The top surface area of the nanowires has been calculated by using the following formula:*A* = *πr*^2^(2)

Figure 3a shows the histogram representation of diameter distribution of pristine ZnO nanowires, the average diameter range is around 100 nm. The ionic radii for Zn^+^ and Li^+^ are 0.74 Å, 0.6 Å, respectively, which is why the Li-doped ZnO nanowires have not shown any significant change in the diameter range (i.e., around 100 nm range). Na [32] being the second element in Group-I elements has a larger atomic radius and it has eleven electrons in total and one valance electron. Although the Na^+1^ ion radius is smaller than in the Na atom since one electron has been removed, even then Na^+1^ has ten electrons in two energy levels so the diameter range of Na-doped ZnO nanowires is slightly changed and it is around 105 nm. Similarly, K [33] being third element in Group-I elements having larger atomic radius with 19 electrons. Electronic configuration for K is 1s^2^ 2s^2^ 2p^6^ 3s^2^ 3p^6^ 4s^1^ having four energy levels, even though K^+1^ ion has 18 remaining electrons in three energy levels but still big ionic radius, so the diameter range (110 nm) for K-doped ZnO nanowires has been increased further. As the doping concentrations were minute, there was not a drastic change in all three cases.

Figure 4 shows the XRD pattern of doped and un-doped ZnO nanowires. Obtained peaks were matched by standard (JCPDS Card no 36–1451) hexagonal wurtzite structure peaks. Diffracted pattern indicated the growth orientation of ZnO nanowires along crystal planes (100), (002), (102), and (110). However, the prominent peak along (002) plane confirmed preferential growth orientation of most of the nanowires were along the c-axis. A little shift of the peaks towards the lower angles indicated the minute doping concentrations of group I elements. Shifting of the peaks was due to interstitial defects or due to small dislocations created by the dopants [34]. This could be due to internal stress after doping. It was quite encouraging to observe that there were no secondary peaks in the XRD pattern of all three cases. The XRD pattern revealed that doping concentrations were minute, dopants have been well incorporated in ZnO nanowires, and there was no drastic change in the crystal structure. A dominant peak along the (002) plane indicates the vertical orientation of ZnO nanowires before and after doping. Incorporation of small impurities is well depicted by small peaks in XPS results. Figure 4d–f shows doping concentration of Na, K and Li in ZnO nanowires. Concentration mapping has been plotted by using the Scherrer-Debye equation [35] in which D is the crystallite size (nm), K (0.9) is Scherrer constant λ is the wave length (0.15406 nm) of the source used, β is FWHM (full width at half maximum) in radians, and θ is the peak position in radians.
D = K λ/β cos θ (3)

Mapping concentration shows the uniform distribution of group I elements in ZnO nanowires. Diffracted domain size has been estimated as a function of doping concentration of dopants. A change in diameter after doping has been plotted in Figure 4d–l, as we have discussed earlier that variation in the diameter range after the doping has been due the different ionic radii of the dopants. The mapping concentration shown in Figure 4 is consistent with results shown in Figure 3.

XPS spectra for Zn and O is shown in Figure 5a,b. The two peaks at BE 1020 eV and 1040 eV are related to Zn 2p_3/2_ and Zn 2p_1/2_ states, respectively. Figure 5b shows the oxygen spectra in which peaks at BE at 529.6 eV and at 532.2 eV are attributed to O presence in ZnO. High resolution XPS-spectra has been analyzed to elucidate the presence of Na, Li and K in ZnO nanowires. Na1s XPS spectra has been deconvoluted in two peaks, and two overlapping peaks at binding energy 1070.9 eV, 1071 eV have been merged to one prominent peak at 1070.9 eV which has been deconvoluted. Both Na^+1^ ion peaks attributes Na1s states as shown in Figure 5e. It confirms the successful incorporation of Na in ZnO lattice, obtained data is in good agreement with previously reported values [36]. Similarly, peaks at 295.4 eV and 292.7 eV are attributed to K2p_3/2_ and K2p_1/2_ states, respectively, as shown in Figure 5d. K peaks elucidated formation shallow acceptor levels in ZnO nanowires, earlier [37] Li et al. has studied the role of K in Li ion battery and also found the peaks in the same range. Shallow acceptor levels support *p-type* doping and supposed to generate more holes than deep acceptor levels. K^+1^ ion has an electronic configuration of 1s^2^ 2s^2^ 2p^6^ 3s^2^ 3p^6^ corresponds to a stable state than K (1s^2^ 2s^2^ 2p^6^ 3s^2^ 3p^6^ 4s^1^). It can be seen that K^+1^ ion has eight electrons in the third energy level (3s^2^ 3p^6^), i.e., completely filled state while K due to 4s^1^ is highly unstable, so K^+1^ is supposed to form stable acceptor levels in ZnO nanowires. These shallow acceptor levels have captured the free moving electrons in ZnO nanowires and reduced the leakage current through ZnO nanowires, enhancing the output voltage. Similarly, there is a peak which appears at 53.6 eV corresponding to the core level of Li1s. As Li being 1st element of group I is quite light and small, this is why it could not be easily detected by common characterizations such as energy dispersive spectroscopy (EDS) [38]. It has only three electrons (2 core electrons, 1 valance electron). High resolution XPS-spectra has confirmed incorporation of Li in ZnO structure. The substitutional defect denoted by Li_Zn_ has generated a shallow acceptor level that is ideal for *p-type* doping [39]. This shallow acceptor level has enhanced the internal resistance of ZnO nanowires which consequently raised the piezoelectric output voltage. Moreover, theoretical studies [40] have also predicted Li_Zn_ substitutional defect is more stable than Li interstitial (Li*_i_*), Li*_i_* is supposed to highly mobile in ZnO. Li^+1^ ion is stable due to its 1s^2^ configuration just as with He (Noble gas). Li_Zn_ has improved the piezoelectric performance of the nanogenerator which could be observed in Figure 6c–e Output voltage has shown significant rise as the doping concentration was increased.

Figure 6a confirms the generation of piezoelectric potential voltage and its deliverance at output stages. Figure 6b shows the switching polarity test which ensures the voltage generation is truly from VING structure but opposite to spectrum shown in Figure 6a (i.e., due to switching of terminals). Schottky contact [7,8,20] is a very crucial factor in piezoelectric nanogenerators; it ensures the flow of current in one direction only and prevents the flow of current through the nanowires. Figure 6a,b confirmed the formation of Schottky contact in between ZnO nanowires and the upper electrode. Figure 6c–e shows output voltage generated by K-doped ZnO nanowires. Potassium has an electron affinity of 0.501 eV and it has been clearly manifested in voltage graphs, voltage has been gradually enhanced as the doping concentration is increased from 2% to 6 wt.% There are two pertinent reason of voltage rise is. One is the creation of shallow acceptor level as depicted in XPS spectra, and the second reason is the electron affinity of the dopants. Shallow acceptor levels have generated holes in the ZnO nanowires which trapped free moving electrons. Free moving electrons tend to deplete the piezoelectric potential [20] and cause reduction in the output voltage of nanogenerator. In other words, reverse leakage current through nanowires have been reduced by an electron trapping mechanism. Conduction of nanowires is associated with quantum conductance “G” that is integral multiple of the factor e^2^/2 h [12]. In K-doped ZnO nanowires conducting channels were reduced and leakage current has been cut down. Figure 6f shows column bar graph indicating the voltage rise to corresponding doping concentration of K.

Figure 7a represents periodic piezoelectric voltage generation by pristine ZnO nanowires and Figure 7b confirms the voltage occurrence range. Figure 7a confirmed that the Schottky contact has been established in between ZnO nanowires and Au electrode and histogram manifested the voltage range for both and negative cycles. The electron affinity for Na is 0.547 eV slightly higher than K and output voltage produced by Na-doped ZnO is higher than K-doped ZnO nanowires as shown in Figure 7c–e. The max voltage produced by 6% K/ZnO nanowires is 1.95 V and same concentration of Na has produced an output voltage of 2.34 V. The voltage bar graph shown in Figure 7f depicted the obtained output voltages corresponding to the doping concentrations of Na. Na has an electronic configuration of 1s^2^ 2s^2^ 2p^6^ 3s^1^ which is quite unstable due to 3s^1^ while Na^+1^ ion has stable configuration 1s^2^ 2s^2^ 2p^6^. Na^+1^ produces stable acceptor levels as indicated by XPS results. Na^+1^ ion has high tendency to capture free electrons. It has been observed that as the doping concentration is increased the output voltage is enhanced. Rise in concentration of Na^+1^ ions have captured more free electrons and reduced leakage current through ZnO nanowires. The effect has been well studied through output voltage graphs and these graphs indicate that voltage has been gradually increased. Previously, Na-doped ZnO nanowires have been mostly used for photocatalytic applications [24,34]. The role of Na-doped ZnO nanowires have also been investigated for nanogenerators. Earlier, researcher claimed that Na has produced interstitial defects and enhanced the adsorption phenomenon and luminescence properties were improved and also reduced organic pollutants from clean water [41,42,43], but recent findings were quite unique in suggesting that Na^+1^ ions were used to mitigate the leakage current through ZnO nanowires. The obtained results were quite encouraging.

Figure 8a represents the piezoelectric voltage generation by un-doped ZnO nanowires and Figure 8b shows voltage occurrence range for positive and negative voltage. The electron affinity for Li is 0.618 eV, which is greater than both K and Na, which is why Li-doped ZnO nanowires have produced maximum voltage peaks. As discussed earlier, the Li ionic radii are comparable to Zn and its tendency to accept free electrons is high; Li-doped ZnO nanowires have produced their maximum output voltage as compared to other Group-I elements. Li has three electrons in total including two core electrons and one valance electrons. The valance electron is being protected by two core electrons. The effective nuclear charge on Li cation is Z_eff_ = 3 − 2 = +1 and shallow acceptor level formed by Li_Zn_ trapped free moving electrons more than the other cations (Na^+^ and K^+^) which consequently minimized reverse leakage current through ZnO nanowires and enhance piezoelectric output voltage. The other reason of high output voltage generation could be the rise in the piezoelectric coefficient. Replacement of Zn ion by Li ion has not only raised the internal strain of ZnO lattice but also raised anisotropy of the ZnO wurtzite structure and both factors contributed in raising the piezoelectric coefficient [27]. Chang et al. [44] studied piezoelectric response of in Li-doped ZnO nanowires by using AFM. A Pt coated Silicon tip has been used in contact mode to measure the output current, 8 nA and output voltage of 160 mV. In a recent study, the generated voltage has been enhanced by first using acceptor impurities of group I elements and, afterwards, doped ZnO nanowires have been oxidized (with O_2_) to trap the remaining free carriers. The two-step approach has shown a considerable rise in the resulting piezoelectric output voltage. Ashfiqual [45] reported the role of Li-doped ZnO nanowires in gas sensing and energy harvesting applications. The fabrication process was quite complex, pulsed laser deposition (PLD) technique has been used for the deposition of insulation layer, adhesion layer, and lift-off process is also being used, as well as special template, photoresist, wet etching, etc. In another recent study, facile aqueous route has been opted for the synthesis of pristine ZnO and doped ZnO nanowires and furthermore, ZnO nanowires have been modified by O_2_ molecules to mitigate the screening effects. It is a very cost-effective easy procedure to fabricate high output voltage devices. Theoretically [41], it has been predicted that when Li ion enters the ZnO crystal it replaces Zn to generate Li_Zn_ which increases the strain and anisotropy of crystal which overall increases the piezoelectric coefficient. Intrinsically, ZnO nanowires are *n-type* due to oxygen vacancies and Zn interstitials so by doping Li as *p-type* impurity has generated more holes inside ZnO nanowires that reduced the conductivity of the nanowires. Quantum conductance G of nanowires is closely dependent on available conducting electrons while free electrons have been trapped by the holes the conducting channels were also been reduced and internal resistance of ZnO nanowires has been increased. As the leakage current is reduced, consequently the piezoelectric output voltage has been raised (as shown in Figure 8c–e). Figure 8f depicts the voltage bar graph, indicating gradual voltage rise with respect to the doping concentrations of Li. Earlier studies [46] have also pointed out the role of Li-doped ZnO nanowires for dilute magnetic semiconductors (DMS) which could be due to Zn vacancy. Studies predicted that ferromagnetism was probably due to Zn vacancy and not due to the neutral oxygen vacancy. Theoretical calculations have predicted that Zn vacancies are responsible for tiny magnetic moments (1.3 µB) in host semiconductors. Although Li is termed as non-magnetic but the creation of Zn vacancies in Li-doped ZnO system has originated ferromagnetism [47] but recent study has been focused on generating high output voltage by increasing the number of holes in ZnO nanowires.

Gao. Y et al. [48] theoretically predicted that screening effects is one of the major issues in piezoelectric nanogenerators. By reducing screening effects, the piezoelectric voltage could be enhanced significantly. The screening effect phenomenon is due to free moving electrons which are being attracted towards positive side of the electric dipoles during external stress. Electric dipoles developed due to external stress are crucial for piezoelectric voltage generation and screening effects tend to deplete it. Yang et al. [49] experimentally verified the screening effect phenomenon and also claimed a tunneling of electrons during the bending of the nanowires which depleted piezoelectric potential. It has been established in our earlier reports [15,16,17,18] that by surface modification of ZnO nanowires screening effects could be minimized considerably. Adsorption of O_2_ molecules on the surface defects of ZnO nanowires created an ionic layer that acted as barrier layer for electrons to jump through the nanowires.
O_2_ + e^−^ → O_2_^−^
(4)

The effect has also been witnessed by Xue et al. [50] when he applied same external force on ZnO nanowire placed in dry air and one placed in pure oxygen environment and 0.3 V has been raised in pure oxygen gas environment. This is due to the mitigation of screening effects. The outer layer also stops the electron tunnel between the nanowires during continuous external straining. Adsorbed O_2_ molecules have been produced which pronounce the edge effect as well. Edge effects affect the conductivity of the nanowires since the atoms laying on the surface of nanowires are not being bounded by the adjacent atoms just as in case of bulk in which atoms are surrounded my many other neighboring atoms [51]. There are more surface atoms than inside nanowires which lead to poor conductivity of nanowires. The effect became more prominent in the recent study as O_2_ molecules adsorbed on the outer surface of nanowires. There would be more scattering inside nanowires due to which internal resistance of the nanowires has been increased [17]. Quantum conductance (G = 2e^2^/h) of the nanowires depends upon the available number of conducting channels, in which “e” is the charge on the and electron “h” is the Planck constant. Furthermore, the oxide layer worked as barrier for the electrons to tunnel through the nanowires and the screening effect is also minimized. The effect of all above mentioned factors has given rise to a certain piezoelectric output voltage. Figure 9a–c explained it well that oxidized nanowires generated high voltage peaks than the un-oxidized nanowires. Open circuit output voltage for VING has been monitored by gradually changing the load resistance from (2–12 MΩ) as shown in Figure 9d. A maximum output voltage of 4.47 has been recorded. Figure 9e clearly indicated the generated voltage difference between oxidized and un-oxidized VING.

Z L Wang [13] used a LING (lateral integrated nanowire generator) structure to generate 20–25 mV. Similarly, a ZnO nanowires-based LING attached to the body of running hamster has produced maximum voltage of 100 mV [51]. Wang also designed a woven nanogenerator [52] that works the friction produced during the rubbing of cloths. He used two different fibers, namely one covered by ZnO nanowires and another fabric coated by a palladium (Pd) layer. Pd worked as Schottky contact, and during the rubbing of the cloths frictional energy was converted into useful electrical energy. It generated a tiny voltage of 3 mV. Sheng et al. reported a maximum of 1.26 V by using the LING (lateral integrated nanowire generator) configuration [12]. The LING configuration has some limitation. There is always a chance that during a continuous external straining, nanowires pluck out from the electrodes and also from the substrate (due to which the output voltage gets disrupted). This requires more fabricating steps, and sometimes it requires masking pattern electrode positioning and growth of nanowires. Afterwards, liftoff procedures or sometime etching for removal of unwanted portions makes LING a bit of a cumbersome configuration. While VING is a quite hassle-free configuration, our recent study is continuation of that. It does not require any complex fabricating steps such as lithography, masking, etching, and moreover it is a very cost effective technique. Recent results are quite competitive and very good addition of the to the tally of nanogenerators. Nanogenerators hold the key of many future problems and could play a pivotal role in future applications such as in electric vehicles, smart watches, smart keyboards, smart computer mouse, smart health care systems, etc. These tiny NGs can generate any desired levels of high voltages by employing them in serial combinations. In the recent case, by combining three NG in series more than 12 V could be achieved. Proposed mechanical transducers could be good replacement of the batteries from future devices.

## 4. Conclusions

We have successfully grown *p-type* doped ZnO nanowires by using a facile aqueous route. Group I (K, Na and Li) doped ZnO nanowires have generated high piezoelectric voltages. XPS results indicated the presence of desired acceptor impurities. A slight shift the XRD pattern of doped ZnO nanowires was due crystal strains produced by the dopants. XPS and XRD findings are being well accredited by high output voltage spectra of doped ZnO nanowires. Pertinent reason behind the of high piezoelectric voltage generation is due trapping of free electrons by the shallow acceptor levels produced by group I dopants. The second salient feature of the study was to mitigate the screening effects by using surface modified ZnO (oxidized with O_2_) nanowires. Adsorbed oxygen molecules not only reduced the screening effects but also hampered the tunneling of electrons during an external stress. Maximum output voltages generated by K/ZnO nanowires, Na/doped and Li/ZnO nanowires were 1.95 V, 2.34 V, and 3.54 V, respectively, and for surface modified *p-type* doped ZnO nanowires voltages were further enhanced to 2.48 V, 3.19 V, and 4.47 V, respectively. Proposed VING is highly recommended for self-powered Micro/Nano scale devices.

## Figures and Tables

**Figure 1 micromachines-13-01450-f001:**
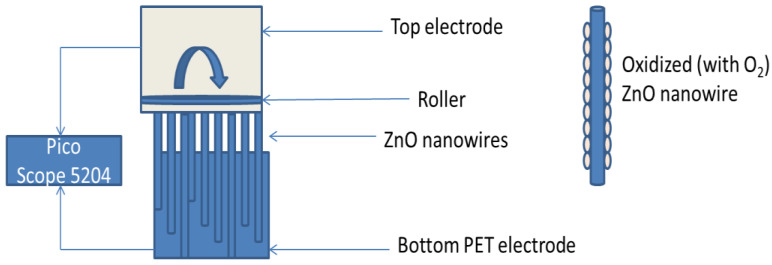
Schematic diagram of ZnO based VING and oxidized ZnO nanowire.

**Figure 2 micromachines-13-01450-f002:**
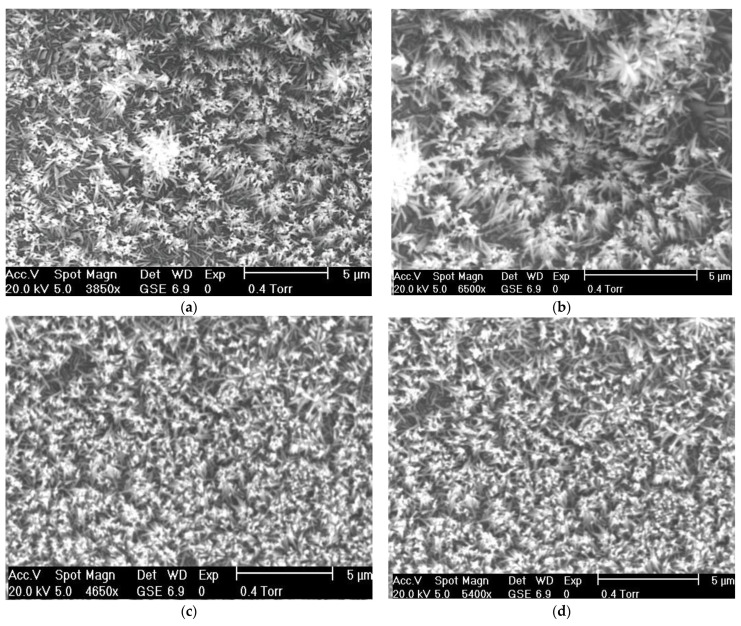
SEM images of ZnO nanowires, (**a**) pristine ZnO nanowires, (**b**) Li-doped ZnO nanowires, (**c**) Na-doped ZnO nanowires, and (**d**) K-doped ZnO nanowires.

**Figure 3 micromachines-13-01450-f003:**
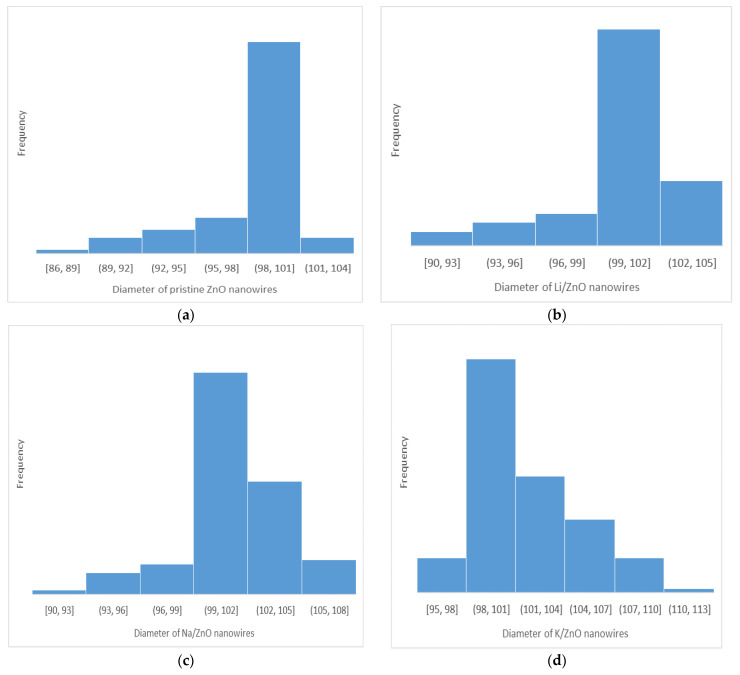
Histogram showing diameter range of ZnO nanowires, (**a**) pristine ZnO nanowires, (**b**) Li-doped ZnO nanowires, (**c**) Na-doped ZnO nanowires, (**d**) K-doped ZnO nanowires.

**Figure 4 micromachines-13-01450-f004:**
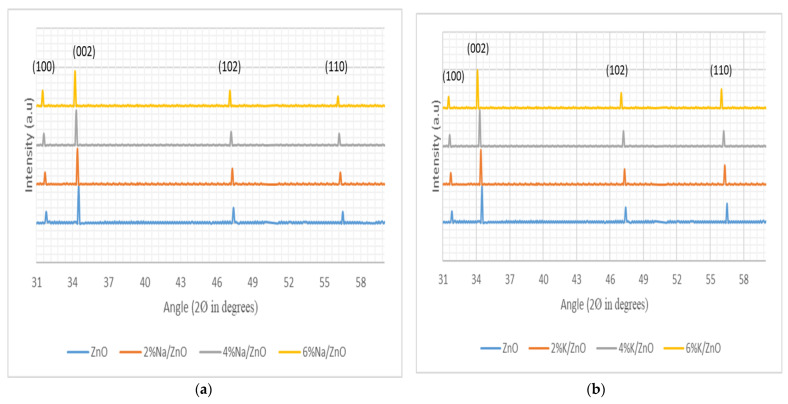
XRD pattern of vertically grown ZnO nanowires, (**a**) Na-doped ZnO nanowires, (**b**) K-doped ZnO nanowires (**c**) Li-doped ZnO nanowires, and (**d**–**l**) concentration mapping of doped Li, Na, and K, respectively.

**Figure 5 micromachines-13-01450-f005:**
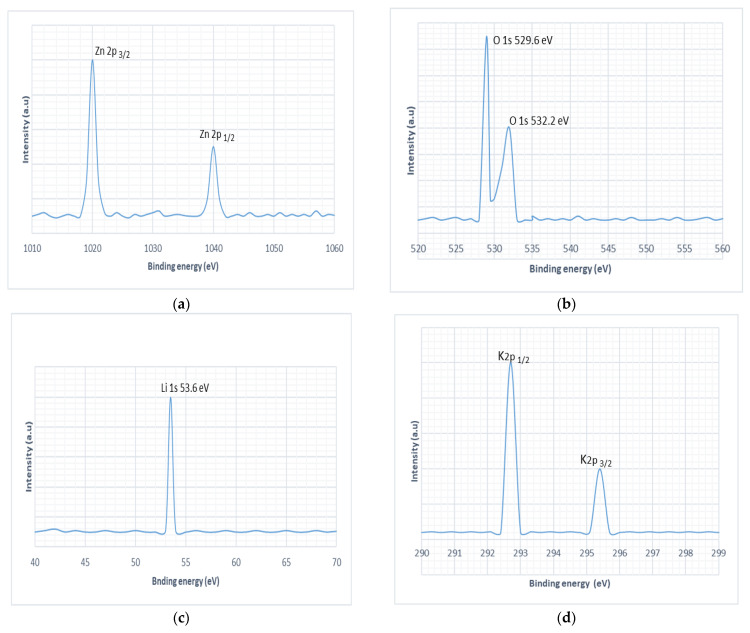
XPS spectra, (**a**) Zn 2p (**b**) O 1s, (**c**) Li 1s, (**d**) K 2p, and (**e**) Na 1s.

**Figure 6 micromachines-13-01450-f006:**
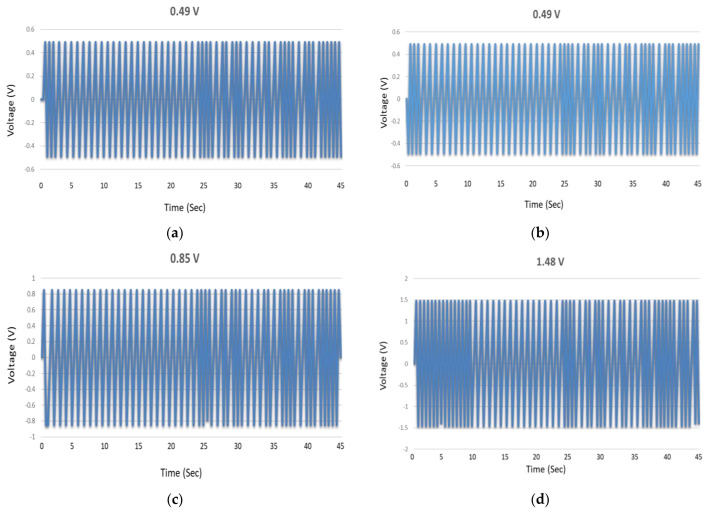
Output voltage values generated by ZnO nanowires based VING, (**a**) as- grown, (**c**) 2%K/ZnO (**d**) 4% K/ZnO (**e**) 6% K/ZnO (**b**) voltage histogram, (**f**) voltage bar graph for various K doping concentrations.

**Figure 7 micromachines-13-01450-f007:**
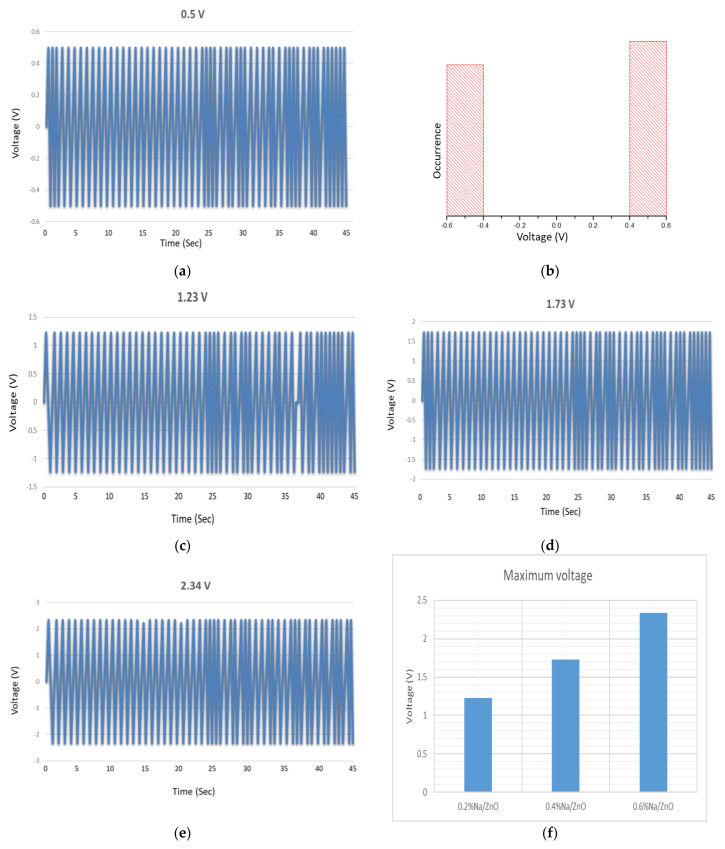
Output voltage values generated by ZnO nanowires based VING, (**a**) as-grown, (**c**) 2%Na/ZnO (**d**) 4% Na/ZnO (**e**) 6% Na/ZnO (**b**) voltage histogram, (**f**) voltage bar graph for various Na doping concentrations.

**Figure 8 micromachines-13-01450-f008:**
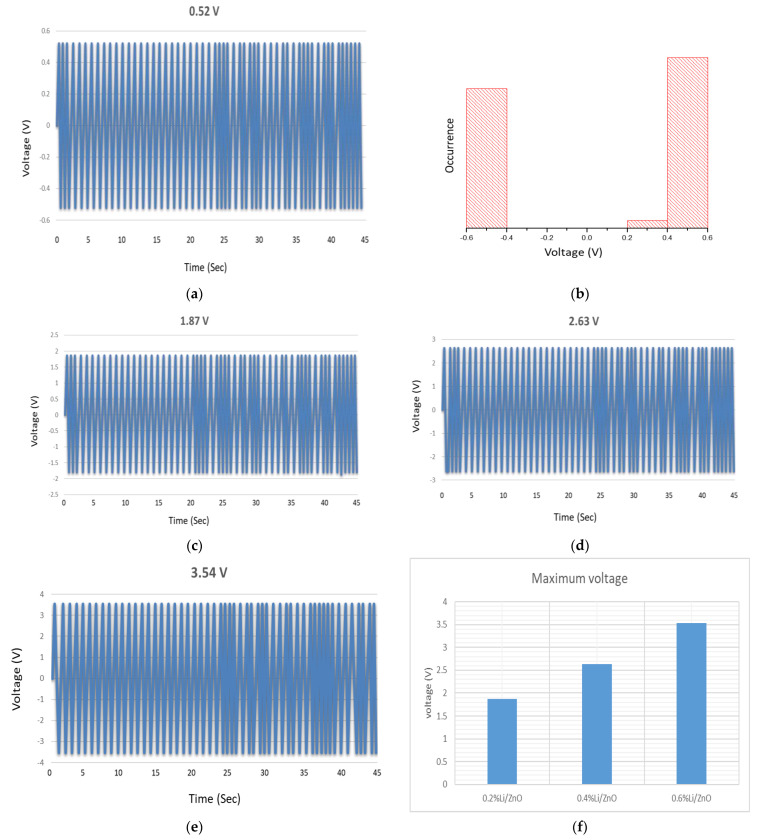
Output voltage values generated by ZnO nanowires based VING, (**a**) as-grown, (**c**) 2% Li/ZnO (**d**) 4% Li/ZnO (**e**) 6% Li/ZnO, (**b**) voltage histogram, and (**f**) voltage bar graph for various Li doping concentrations.

**Figure 9 micromachines-13-01450-f009:**
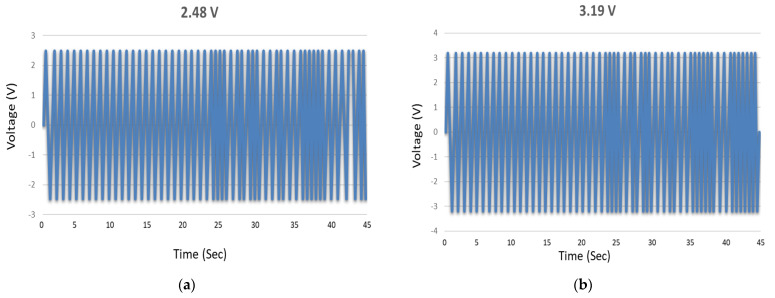
Output voltage values generated by ZnO nanowires based VING oxidized for 5 h, (**a**) 6%K/ZnO, (**b**) 6%Na/ZnO, (**c**) 6% Li/ZnO, (**d**) voltage increment by changing load resistance, and (**e**) voltage comparison for all three cases with and without oxidation.

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
