# Peer review of "Effects of Group-I Elements on Output Voltage Generation of ZnO Nanowires Based Nanogenerator; Degradation of Screening Effects by Oxidation of Nanowires"

_micromachines, 2022, doi:10.3390/mi13091450_

Round 1
Reviewer 1 Report
Authors make a research on the output of ZnO nanowires with doping different group-I elements. By doping these elements, the output of ZnO nanowires has been improved. This idea is reasonable and meaningful. However, some direct evidences are missing. I will make a decision after a major revision.
There is a doubt about the doping ratio. It is totally different in Figures and full text. All the element mappings must be provided to proof the doping ratio. If missing, I will not continue consideration of this manuscript.
Here are some other comments.
- In the abstract, the authors should explain the reason why doping K, Na, Li elements improves the output.
- Please provide the 90°-section SEM of ZnO nanowires. It is necessary to proof your VING.
- The optical image of VING is also necessary.
- In Figure 3, the axis title should be checked again.
- The SEM of ZnO nanowires with/without doping should be provided. Could doping elements influence the length of ZnO?
- What are the X-axis titles of Figure 4d-f?
- What is the unit of y-axis of Figure 6f?
- Authors should explain how the output of the ZnO nanowires is measured.
- In Figure 9, authors should provide the resistance of ZnO.
- Authors should provide more explanations on why doping K, Na, Li elements improves the output.
- The English should be further checked.
Author Response
Response to reviewer 1 comments.
1-It is being mentioned in the abstract the dopping of Li, Na and K have created shallow acceptor levels which trapped free moving electrons inside ZnO nanowires. Fee moving electrons cause leakage current through the nanowires and deplete piezoelectric potential. Reverse leakage current is a bad factor for piezoelectric nanogenerators. Reduction in reverse leakage current signifies the rise in output voltage.
2-3 I am extremely sorry for 90o section SEM view as SEM machine is not in service due to mantenance issues.
4- Axis of the figures is corrected.
5- SEM image of perstine and doped ZnO nanowires have shown in fig.2 (a-d)
Fig.2 SEM images of ZnO nanowires, (a) pristine ZnO nanowires, (b) Li-doped ZnO nanowires, (c) Na-doped ZnO nanowires, (d) K-doped ZnO nanowires
6- X-axis titles of fig.4(d-f) have been added.
|
|
|
|
|
|
|
|
|
|
(d-l) concentration mapping of doped Na, K and Li respectively
7- Y-axis of the fig.6 (f), fig.7(f) and fig.8(f) is output voltage.
|
|
|
Fig.6 (f) voltage bar graph for various K doping concentrations, Fig. 7 (f), voltage bar graph for various Na doping concentartions, Fig. 8 (f) voltage bar graph for various Li doping concentrations.
8- Output voltage has been measured by using Picoscope 5204 as shown in fig.1.
9-Load resistance has been gradually increased from 2MΩ to 12 MΩ to achieve the optimal output voltage. Rise in piezoelctric voltage with change of load resistance has already been reported earlier [12].
10- Doping of group-I elements is very significant as they behaved as acceptor impurities. Acceptor impurities are supposed to be conducive for ZnO base nanogenerators [13]. Free moving electrons are bad factor for pezolectric potential and acceptor dopants captured free moving elctrons. Due to which reverse leakage current through nanowires have been reduced and consequently output voltage has been increased.
11- English language has been checked.
References:
[12] X. Sheng, Q. Yong, Chen Xu, Yaguang, Z.L. Wang, Nat. Nanotechnol. 46 (2010) 366.
[13] Ming P L, Ming Y L, Lih J Chen, Nano Energy 1 (2012) 247.

Reviewer 2 Report
1) In Fig 4, the Li-doped samples show peaks in lower angles compared to the undoped samples. As I think, the opposite trend is expected to appear in XRD measurements. That is, the Li-doped samples should show peaks in higher angles. I recommend you to take a more close look at your results to avoid any possible mistakes, even if I might be misunderstood.
2) In Fig. 4 (d)-(f), authors are recommended to more clearly explain the relationship between the diffracted diameter size and doping concentration in each figure.
3) In Fig. 5, authors explained that the shallow doping could be determined through the XPS spectra of Li, Na, and K. I am wondering if this argument is true, and I hope that authors could describe their mutual relationships more clearly.
Author Response
Response to Reviewer 2 comments.
- In group I elements, Li is supposed to be an ideal p-type doping candidate due to its smaller ionic radii of (0.76 Ao) which is closest to Zn (0.74 Ao). It can easily substitute LiZn that worked as electron acceptor impurity [45]. Slight shifting of the peaks indicated the internal stress developed during LiZn As the doping concentrations were very minute and no additional peaks were observed in the XRD pattern indicated successful substitution of Li in ZnO matrix. However, the argument of shifting the peaks towards larger angles could be true for heavily doped samples.
2- X-axis titles of fig.4(d-l) have been added.
|
|
|
|
(d-l) concentration mapping of doped Li, Na and K respectively.
3-Interestingly, Li worked as amphoteric impurity in ZnO lattice and produces LiZn by substituting Zn atom. LiZn is supposed to be more stable due to kick-out mechanism [27], in kick-out diffusion mechanism free moving electrons are being trapped by LiZn and Zni2+ ions were produce that shifts the fermi level Ef more towards the valance band Ev, consequently free moving electrons were reduced from ZnO nanowires. It signifies the rise in piezoelectric potential because as the free moving electrons were minimized the reverse leakage current through nanowires was also reduced. Formation of shallow acceptor levels due to kick-out diffusion mechanism is being justified by the core level core peak of Li 1s as shown in fig.5 (c). Similarly, Na 1s two peaks were being overlapped which were being deconvoluted by high resolution XPS. Shallow acceptor level NaZn+ were detected by two overlapping signals. Due to shallow acceptor levels produced by Li and Na they are regarded as better acceptors than group V elements because group I elements (Li, Na, K) substitute Zn-site rather than O-site [36].
References
[27] H.M. Ashfiqul Hamid, Z Çelik Butler, Nano Energy 50 (2018) 159.
[36] A Saaedi, R Yousefi, F Sheini, M Cheraghizade, Ceramic International 40 (2014) 4327.
[45] Chang Yu, Jui Yuan Chen, T P Yang, C W huang, Nano Energy 8 (2014) 291.

Round 2
Reviewer 1 Report
Congratulations! The quality of this manuscrpit has been improved. And I suggest that this paper can be accpeted by Micromachines.